# COVID-19—Awareness and Practice of Dentists in Saudi Arabia

**DOI:** 10.3390/ijerph18010330

**Published:** 2021-01-05

**Authors:** Bassel Tarakji, Mohammad Zakaria Nassani, Faisal Mehsen Alali, Anas B. Alsalhani, Nasser Raqe Alqhtani, Abdullah Bin Nabhan, Adel Alenzi, Ali Alrafedah

**Affiliations:** 1Department of Oral and Maxillofacial Surgery and Diagnostic Sciences, College of Dentistry, Prince Sattam Bin Abdulaziz University, Al Kharj 16245, Saudi Arabia; oralpathofaisal@gmail.com (F.M.A.); n.alqhtani@psau.edu.sa (N.R.A.); a.binnabhan@psau.edu.sia (A.B.N.); a.alenazi@psau.edu.sa (A.A.); a.alrafedah@psau.edu.sa (A.A.); 2Department of Restorative and Prosthetic Dental Sciences, College of Dentistry, Dar Al Uloom University, Riyadh 11512, Saudi Arabia; mznassani@hotmail.com; 3Department of Oral Medicine and Diagnostic Sciences, Alfarabi College of Dentistry and Nursing, Riyadh 11691, Saudi Arabia; dr.anas_salhany@hotmail.com

**Keywords:** COVID19, infection control, dental practice, prevention, awareness, practice

## Abstract

Dental professionals have a major role in the fight against the spread and transmission of COVID-19. This study aimed to evaluate awareness and practice of dentists in Saudi Arabia regarding COVID-19 and the utilization of infection control methods. A 24-item questionnaire was developed and distributed through social media to 627 dentists working in Saudi Arabia. 177 questionnaires were completed (28.2% response rate). Most dentists were aware about the transmission, incubation time and main clinical symptoms of COVID-19. Almost 83% of the respondents appreciate the risk of droplets, aerosols and airborne particles in transmission of COVID-19 in the dental clinic. Among the common practices of participants are measuring patient’s body temperature before undertaking a dental treatment (88.7%), cleaning the environmental surfaces at the dental clinic after each patient (91.5%) and restriction of dental treatment to emergency cases (82.5%). It seems that practicing dentists in Saudi Arabia are fairly aware about COVID-19. The practices of the surveyed dentists appear to be mostly consistent with the current guidelines and recommendations for infection control of COVID-19 in the dental clinic. Some drawbacks in knowledge and a number of inappropriate practices can be identified and require the attention of health authorities.

## 1. Introduction

COVID-19 is a respiratory disease that has rapidly spread through the whole world and turned into a pandemic. It is caused by a novel coronavirus and was first detected in December 2019 in Wuhan, China [1,2]. The World Health Organization (WHO) announced COVID-19 as a global disease and called for an international collaboration to overcome this health crisis [3]. Research findings indicated that COVID-19 can spread from person to person via respiratory droplets through coughing, sneezing, or even talking; by physical contact with an infected person; or by touching a contaminated surface [4]. The main clinical symptoms of COVID-19 are fever, cough and shortness of breathing [4]. In dental practice, the risk of transmission of COVID-19 is quite high due to inevitable close contact between dentists and their patients [5]. As a preventive measure, D’Amico et al. [6] recommended not having multiple patients in the waiting room and maintaining a distance of at least 1.5 m between persons.

Airborne droplets and aerosols have been identified to be the main route for transmission of COVID-19 in the dental clinic [5]. Unfortunately, most types of dental intervention involve droplet/aerosol generating procedures and this shows the high level of risk for infection with COVID-19 in the dental environment. Although patients infected with COVID-19 are usually denied dental treatment, asymptomatic COVID-19 patients may pass the screening procedures at the dental clinic and could easily spread the virus to dental staff and attending dental patients [7]. Telemedicine, tele-dentistry and smart phones are important tools to prevent virus transmission and to perform a quick diagnosis and management at medical/dental offices [8]. Another challenge for dentists is the potential for reactivation of COVID-19, and this may have an implication on the right time to offer dental treatment for patients who have recently recovered from the infection [9,10]. Reactivation of COVID19 implies that the recovered patient may still carry the virus and an extra round of viral detection and isolation may be required [10]. Such patients can still carry the viral load and transmit the virus even after they have tested negative following 14 days from onset of symptoms and isolation [10].

So far, guidelines and recommendations to control the spread of COVID-19 have been issued by the American Dental Association, the World Health Organization and other health authorities across the world. These recommendations include assessment of patients before treatment, hand washing, use of personal protective equipment, use of rubber dam, mouth rinsing before dental procedures, use of anti-retraction handpieces, and other measures [11,12,13,14]. Despite the good news of the release of an effective vaccine against COVID-19, COVID-19 still exists, with no consensus on antiviral treatment. A second wave of the pandemic has also forcefully started in many countries [15]. Furthermore, it is contemplated that much time is needed before a vaccine is in reach of every country in the world. Therefore, applying preventive measures to control COVID-19 infection can be considered the most critical approach at the current time. Evidently, the role of dental professionals in applying such preventive measures is of major importance and any negligence/ignorance from their side may result in disastrous transmission of the deadly virus. Even with the availability of prevention guidelines on disease control, there is a need to shed some light on what is going on in clinical practice. Examination of dentists’ awareness and practice regarding coronavirus, its transmission routes and disinfection procedures can be considered the first step in underlining/managing any misunderstanding or inappropriate actions in daily dental practice. This is to ensure safe dental treatment for our patients and dental staff alike. The aim of this survey was to evaluate awareness and practice of dentists in Saudi Arabia regarding COVID-19 and of utilized infection control methods.

## 2. Materials and Methods

### 2.1. Study Design

A cross-sectional descriptive study.

### 2.2. Setting and Sample

The target population of this research were practicing dentists in Saudi Arabia.

### 2.3. Instrument

Based on a review of the current literature, a questionnaire to assess awareness and practice of dentists regarding COVID-19 was developed by the authors. The questionnaire was first piloted among 20 dentists active in social media and working in different areas in Saudi Arabia to examine validity of the items and clarity of the provided information. The collected remarks were reviewed and the questionnaire items were modified accordingly. The final version of the questionnaire was then prepared and distributed among the targeted dentists. The structure of the questionnaire comprised four parts; a cover page, demographic questions, eight statements related to COVID-19 to assess dentists’ awareness, and the presentation in the last part of 16 questions related to practice at the time of the coronavirus pandemic. English was the language of choice to present the questionnaire items. This is to attract practicing dentists from different nationalities to take part. The ethical approval for this study was obtained from Prince Sattam Bin Abdulaziz University (Approval No: REC-HSD-026-2020).

### 2.4. Data Collection

Google forms were used to make an electronic copy of the questionnaire. The questionnaire was then distributed through social media to a random sample of dentists working in Saudi Arabia. The purpose of the survey was explained and dentists were invited to complete the questionnaire. Dentists were assured about confidentiality and anonymity of the collected data. More than a four-month timeline (June to September 2020) was allowed to collect data and over this time frequent reminders were sent to dentists to complete the survey items.

### 2.5. Data Analysis

The SPSS statistical package was used for data analysis (IBM SPSS Statistics for Windows, Version 20.0, Released 2011, IBM Corp, Armonk, NY, USA). Descriptive statistics presented characteristics of participating dentists, and frequency tables were generated to illustrate response of dentists to survey questions. The Chi-Square statistic was used to assess any possible association between questionnaire items and dentists’ clinical experience/qualification. A *p*-value < 0.05 was considered significant.

## 3. Results

A total of 627 dentists were contacted in the first round, a second and third reminder were sent to non-respondents, and 177 questionnaires were completed (28.2% response rate). The study population comprised Saudi and non-Saudi dentists of both genders and different ages. Participation covered the various geographical regions of Saudi Arabia with a majority from the central region of the Kingdom (71.2%). Most respondents were working in the governmental sector (67.2%) and 26% were practicing in private dental centers. The clinical experience of the surveyed dentists ranged between 1 and 33 years with a high proportion of dentists with clinical experience between 1 and 5 years (45.2%). More than half of the participants were general dental practitioners (52%), while 48% were specialist dentists (48%). Characteristics of the study population are summarized in Table 1.

Response to statements about COVID-19 indicated that the vast majority of dentists are aware about transmission, incubation time and the main clinical symptoms of COVID-19 (94.9%, 91.5% 96.6%, respectively). Most of the surveyed dentists were also aware of the high mortality rate among COVID-19 patients with chronic disease and the role of isolation of infected people in the reduction of the spread of coronavirus (89.8%, 88.1%, respectively). Almost 83% of the respondents appreciate the risk of droplets, aerosols and airborne particles in transmission of coronavirus in the dental clinic. A considerable proportion of the dentists were not aware that COVID-19 can stay on surfaces for a few days and that RT-PCR is the best test to detect COVID-19 infection (36.7%, 33.9%, respectively). The results are illustrated in Table 2.

Table 3 shows response of participating dentists to practice related questions about COVID-19. It can be noted that a number of practices are common among the vast majority of dentists in this study. This includes updating knowledge with current WHO guidelines for cross-infection control regarding COVID-19 (84.2%), measuring patient’s body temperature before undertaking a dental treatment (88.7%), cleaning the environmental surfaces at the dental clinic after each patient (91.5%), washing hands before and after treatment of every patient (96.6%), restriction of dental treatment to emergency cases (82.5%), and daily body temperature measurements before starting work in the dental clinic for the whole working staff (81.9%). A majority of the responding dentists also indicated further popular practices at the time of the coronavirus pandemic: asking every patient about their travel history before performing a dental treatment (75.1%), delay of the dental treatment for patients showing suspicious symptoms related to COVID-19 (78%), a belief that a N-95 mask should be worn in dental practice rather than an ordinary mask (74%), and contacting the Health Authority to inform about suspicious COVID-19 patients attending the dental clinic (71.8%). On the other hand, a negative response was observed among a considerable proportion of the dentists regarding availability of a screening room at their dental office (33.3%) and use of the rubber dam to minimize aerosols and risk of infection with COVID-19 (31.1%). Similarly, a high proportion of the study dentists responded negatively regarding use of anti-bacterial mouthwash before dental treatment (48%), use only of low speed handpieces to minimize aerosols and risk of infection with COVID-19 (61%), and use of anti-restrictive valve dental handpieces (56.4%). In terms of charging patients for the extra-infection control procedures, 52% of the dentists confirmed that they did so.

When chi-square statistics were used to evaluate any association between questionnaire items and dentists’ clinical experience, only three significant differences were identified. Dentists with greater clinical experience showed significantly higher level of awareness about the role of droplets, aerosols and airborne particles in transmission of COVID-19 in the dental clinic (*p* = 0.014). The practice of senior dentists was also significantly better in terms of measuring patient’s body temperature before provision of a dental treatment (*p* = 0.023) and cleaning the environmental surfaces after each patient (*p* = 0.026). Almost no association was recorded between type of dentists’ qualification and level of awareness/clinical practices at the time of COVID-19 pandemic. Table 4 presents the aforementioned findings.

## 4. Discussion

The eruption of the COVID-19 pandemic has exerted great pressure on workers in the health sector and development of measures to control transmission of the virus has become a priority. Although it is the responsibility of health authorities in any country to provide health professionals with guidelines and orientation on the best practices at the time of a health crisis, health professionals in their turn should adhere to any provided guidelines or recommendations. This is clearly a sign of standard medical care that gives attention to the health and safety of the whole community. The current survey comes in this context as it aimed to evaluate the current awareness and practice of dentists working in Saudi Arabia regarding coronavirus and adopted methods for infection control.

Despite the low response rate in this survey, it is comparable to a similar recent Saudi survey by Shahin et al. [16] who reported a response rate of 21.7% among dentists about COVID-19 pandemic in Saudi Arabia.

Albaker et al. [17] indicated that the number of dentists working in Saudi Arabia was 16,887. Most of them professionally registered as general dentists (70.27%). The percentage of general dentists among professionally registered female dentists is significantly higher than among their male counterparts (79.71% versus 64.80%). Only 22.08% of the dentists working in the kingdom are Saudi. Over 80% of the Saudi dentists are working in the regions of Riyadh, Makkah, and Eastern province. About 66% of the Saudi dentists are working in the public health sector in comparison to only 20.46% of the non-Saudi dentists. The results show that the population of this survey involved junior and senior dentists of both genders and with different clinical experience. It also involved Saudi and non-Saudi dentists, general dental practitioners and specialists, and dentists from the governmental and private sectors. In addition, the surveyed dentists came from the various geographical regions of Saudi Arabia with a majority from the central region including the capital Riyadh. With such a high quality study population, it is more likely that the findings reflect the current infection control practices and level of awareness among practicing dentists in Saudi Arabia regarding COVID-19.

In terms of awareness, this survey has illustrated a fair level of awareness among the surveyed dentists regarding COVID-19 (Table 2). This can be attributed to the extensive efforts that have been made by the Saudi Ministry of Health to educate health workers and laypeople about this pandemic and associated risk of transmission. The Saudi Ministry of Health also published guidelines and imposed strict infection control measures in hospitals and medical centers [18]. Among the important infection control prevention procedures for COVID-19 is cleaning surrounding surfaces after the visit of each patient for the dental clinic [19]. However, there has been uncertainty among a considerable proportion of participating dentists regarding stability of COVID-19 on surfaces over a few days (Table 2). This is in line with the findings of a recent multicounty survey that indicated lack of awareness of the potential for COVID-19 to remain infectious on inanimate surfaces for between three and nine days [20]. Fiorillo et al. [20] indicated that coronaviruses persist in an infectious state on surfaces for several days, even up to nine days. They mentioned that surface disinfection could be performed with 0.1% sodium hypochlorite or 62%–71% ethanol for 1 min.

Although real-time reverse transcriptase-polymerase chain reaction testing (RT-PCR) is the best-known test to detect COVID-19, there was lack of awareness about this fact among a large proportion of the dentists in this study (Table 2). It is accepted that dentists are usually interested in their specialty and might have limited information about viral respiratory infection. However, dentists constitute an essential part of the health workforce, and with such an unprecedented pandemic they should be well educated in order to transfer this knowledge to their patients and take up their role in the fight against this deadly pandemic.

On the level of practice, the outcome of this survey reveals that the practices of this group of dentists are mostly consistent with the current guidelines and recommendations for infection control of COVID-19 in the dental clinic (Table 3). Nevertheless, a number of inappropriate practices can be picked up in the light of this survey. This is mainly related to non-allocation of a screening room at the dental office, non-use of the rubber dam during restorative procedures, non-use of anti-bacterial mouthwash before dental treatment and use of low speed/anti-restrictive valve handpieces. These procedures are preventive measures and usually intended to minimize the spread of coronavirus. It has been shown that use of mouthwashes containing agents with anti-viral activity such as povidone-iodine can be effective against various respiratory viruses [21,22,23]. It is also well-known that rotary instruments generate a large quantity of aerosols/droplets and use of rubber dam and/or low speed handpieces can significantly reduce spread and quantity of aerosols/droplets and hence the risk of infection with COVID-19 [24].

With respect to the role of droplets, aerosols and airborne particles in the transmission of COVID-19 in the dental clinic, the results show an adequate level of awareness among dentists (83.1%). This is in line with the results of a recent Saudi survey that indicated 92% of the participants believed that the mode of transmission of COVID-19 was droplet inhalation [16].

The economic burden of COVID-19 has been heavy on the whole community, especially health providers and patients alike. This is reflected in the current survey as almost half of the participants indicated they have incurred additional charges on patients for the extra-infection control procedures to avoid infection with COVID-19 (Table 3). It is redundant to say that governmental support for the health sector at the time of health crises is needed to avoid any negative sequences on quality of health care and/or the economy of the healthcare entities.

It can be noted that the association between dentist’s clinical experience/qualification and the 24 survey items was very limited (Table 4). This is not surprising as the COVID-19 pandemic is a public health problem rather than an oral health-related one.

It should be kept in mind that at the time of COVID-19 pandemic every patient attending the dental clinic should be considered as potentially infected by the virus, and during all dental procedures up-to-date infection control policies should be implemented. This requires dentists to be continuously updated about the most recent advances in this respect. Overall, this survey has highlighted the current state of awareness towards COVID-19 and practices of dentists in Saudi Arabia regarding infection control prevention procedures. There are clearly some gaps in the level of dental practice that should be considered and rectified by the health authorities in Saudi Arabia for better control of the spread of coronavirus.

A limitation of this study is the relatively low response rate and small sample size. This may affect the generalizability and accuracy of the findings. A further source of concern is the fact that the majority of surveyed dentists are located in the central region of Saudi Arabia. As well, it is important to recognize that the survey items did not cover the various practice/awareness related topics regarding COVID-19 in the dental office, as this pandemic is recent and knowledge about the disease has evolved very quickly. Future larger and more comprehensive investigations are recommended to confirm the findings of this survey.

## 5. Conclusions

It seems that practicing dentists in Saudi Arabia are fairly aware about COVID-19. The practices of the surveyed dentists appear to be mostly consistent with the current guidelines and recommendations for infection control of COVID-19 in the dental clinic. However, some drawbacks in knowledge and a number of inappropriate practices can be identified and require the attention of the health authorities.

## Figures and Tables

**Table 1 ijerph-18-00330-t001:** Characteristics of participating dentists (No = 177).

**Age**	Mean (SD)	33.49 (6.94)
	Minimum	23
	Maximum	58
**Gender**	Male	137(77.4%)
	Female	40(22.6%)
**Nationality**	Saudi	83 (46.9%)
	Non-Saudi	94 (53.1%)
**Type of Practice**	Governmental	119(67.2%)
	Private	46(26%)
	Both	12(6.8%)
**Practice Location in Saudi Arabia**	North Region	7(4%)
	South Region	6(3.4%)
	Centre Region	126(71.2%)
	East Region	14(7.9%)
	West Region	24(13.5%)
**Clinical Experience (years)**	Mean (SD)	7.7 (6.44)
	Minimum	1 y
	Maximum	33 y
	1–5 y	80(45.2%)
	6–10 y	42(23.7%)
	>10 y	40(22.6%)
**Qualification**	DDS/BDS	92(52%)
	Postgraduate Diploma	29(16.4%)
	MSc	30(16.9%)
	PhD	30(16.9%)
	Board Certificate	11(6.2%)
	General Dental Practitioner	92(52%)
	Specialist Dentist	85(48%)
**Specialty**	General Practice	102(57.6%)
	Prosthodontics	16(9%)
	Operative Dentistry	6(3.4%)
	Endodontics	12(6.8%)
	Paediatric Dentistry	6(3.4%)
	Oral Surgery	8(4.5%)
	Periodontics	3(1.7%)
	Oral Medicine	7(4%)
	Orthodontics	4(2.3%)
	Other	13(7.3%)

**Table 2 ijerph-18-00330-t002:** Awareness of participating dentists of COVID-19 (No = 177).

No	Statements about COVID-19	Dentists’ Response *n* (%)		
		True	False	I don’t Know
1	The transmission of COVID-19 is from human to human.	168 (94.9)	8 (4.5)	1 (0.6)
2	The incubation time of COVID-19 is 14 days.	162 (91.5)	10 (5.6)	5 (2.8)
3	The main clinical symptoms of COVID-19 are fever, fatigue, dry cough, and myalgia.	171 (96.6)	3 (1.7)	3 (1.7)
4	COVID-19 can stay on surfaces for few days.	112 (63.3)	55 (31.1)	10 (5.6)
5	Isolation and deferring treatment of people who are infected with the COVID-19 virus are effective ways to reduce the spread of the virus.	156 (88.1)	11 (6.2)	10 (5.6)
6	Real-time reverse transcriptase-polymerase chain reaction testing (RT-PCR) is the best test to detect COVID-19.	117 (66.1)	11 (6.2)	49 (27.7)
7	The mortality rate is more common in COVID-19 patients with chronic disease than others.	159 (89.8)	12 (6.8)	6 (3.4)
8	Droplets, aerosols and airborne particles are more likely to cause transmission of COVID-19 in the dental clinic.	147 (83.1)	8 (4.5)	22 (12.4)

**Table 3 ijerph-18-00330-t003:** Response of participating dentists to practice related questions about COVID-19 (No = 177).

No	Questions	Dentists’ Response *n* (%)		
		Yes	No	I don’t Know
1	Do you update your knowledge with the current WHO Guidelines for Cross-Infection Control regarding COVID-19?	149 (84.2)	21 (11.9)	7 (4)
2	Have you got screening room in your dental office to evaluate the patients regarding COVID-19 before starting dental treatment?	118 (66.7)	43 (24.3)	16 (9)
3	Do you ask every patient about travel history before performing a dental treatment?	133 (75.1)	37 (20.9)	7 (4)
4	Do you measure patient’s body temperature before doing a dental treatment?	157 (88.7)	13 (7.3)	7 (4)
5	Do you postpone the dental treatment of patients showing suspicious symptoms related to COVID-19?	138 (78)	26 (14.7)	13 (7.3)
6	Do you think a N-95 mask should be worn in dental practice rather than an ordinary mask?	131 (74)	35 (19.8)	11 (6.2)
7	Do you ask every patient to rinse his/her mouth with anti-bacterial mouthwash before dental treatment?	92 (52)	75 (42.4)	10 (5.6)
8	Do you ask your assistant to clean the environmental surfaces at your clinic after each patient?	162 (91.5)	12 (6.8)	3 (1.7)
9	Do you wash hands with soap and water/use sanitizer before and after treatment of every patient?	171 (96.6)	3 (1.7)	3 (1.7)
10	Do you use the rubber dam for all/most cases to minimize aerosols and so risk of infection with COVID-19?	121 (68.4)	43 (24.3)	12 (6.8)
11	At time of coronavirus spread, do you use only low speed hand piece for all cases to minimize aerosols and so risk of infection with COVID-19?	68 (38.4)	94 (53.1)	14 (7.9)
12	Do you use antirestrictive valves dental hand pieces at your clinic?	76 (42.9)	70 (39.5)	30 (16.9)
13	At time of coronavirus spread, do you restrict dental treatment to only emergency cases?	146 (82.5)	21 (11.9)	9 (5.1)
14	Do you contact the Authority of Health if you suspect a patient has COVID-19 infection?	127 (71.8)	38 (21.5)	11 (6.2)
15	At time of coronavirus spread, do you measure your body temperature, and the staff working with you every day before starting work in the dental clinic?	145 (81.9)	24 (13.6)	7 (4)
16	Do you charge the patient for the extra-infection control procedures to avoid infection with COVID-19?	92 (52)	73 (41.2)	11 (6.2)

**Table 4 ijerph-18-00330-t004:** Association between survey items and dentist’s clinical experience/qualification.

Statement/Question about COVID-19	Clinical Experience (Years)			
	1–5 y	6–10 y	>10 y	*p*-Value
Droplets, aerosols and airborne particles are more likely to cause transmission of COVID-19 in the dental clinic.				
true	63 (78.8%)	31 (73.8%)	39 (97.5%)	0.014 *
false	6 (7.5%)	1 (2.4%)	0	
I don’t know	11 (13.8%)	10 (23.8%)	1 (2.5%)	
Do you measure patient’s body temperature before doing a dental treatment?				
yes	64 (80%)	40 (95.2%)	39 (97.5%)	0.023 *
no	11 (13.8%)	2 (4.8%)	0	
I don’t know	5 (6.2%)	0	1 (2.5%)	
Do you ask your assistant to clean the environmental surfaces at your clinic after each patient?				
yes	69 (86.2%)	40 (95.2%)	39 (97.5%)	0.026 *
no	10 (12.5%)	0	1 (2.5%)	
I don’t know	1 (1.2%)	2 (4.8%)	0	
The main clinical symptoms of COVID-19 are fever, fatigue, dry cough, and myalgia.	**Qualification**			
	**General Dental** **Practitioner**	**Specialist Dentist**		***p*-value**
*true*	92 (100%)	79 (92.9%)		0.035 *
*false*	0	3 (3.5%)		
*I don’t know*	0	3 (3.5%)		

* Significant difference at *p* < 0.05 as indicated by Chi-Square Statistics.

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
