# Peer review of "COVID-19—Awareness and Practice of Dentists in Saudi Arabia"

_ijerph, 2021, doi:10.3390/ijerph18010330_

Round 1
Reviewer 1 Report
Dear Authors,
Your manuscript is really current and about an interesting topic.
Despite of this, it should be modified before publication.
Please use MDPI authors guidelines.
Please modify affiliation formatting accordingly.
Please remove subsection from abstract.
Introduction section is focused on covid-19 and not on dentistry/dental office or management. Please improve with the use of recent literature:
D'Amico, C.; Bocchieri, S.; Stefano, R.; Gorassini, F.; Surace, G.; Amoroso, G.; Scoglio, C.; Mastroieni, R.; Gambino, D.; Amantia, E.M., et al. Dental Office Prevention of Coronavirus Infection. In Eur J Dent, 2020/12/08 ed.; 2020; 10.1055/s-0040-1715923.
Cervino, G.; Oteri, G. COVID-19 Pandemic and Telephone Triage before Attending Medical Office: Problem or Opportunity? Medicina 2020, 56, 250.
In introduction section please better state the aim of the manuscript
Please in discussion section be more specific on actual anti-covid protocols in dental offices and dental settings. You should consider to add some information about Covid-19 surface persistence in medical settings, and use these informations as starting point for discussion.
You should improve conclusion section with future perspective of this manuscript
Author Response
Reviewer1
Please use MDPI authors guidelines.
Please modify affiliation formatting accordingly.
Done page 1
Please remove subsection from abstract.
Done page 1
Introduction section is focused on covid-19 and not on dentistry/dental office or management. Please improve with the use of recent literature:
D'Amico, C.; Bocchieri, S.; Stefano, R.; Gorassini, F.; Surace, G.; Amoroso, G.; Scoglio, C.; Mastroieni, R.; Gambino, D.; Amantia, E.M., et al. Dental Office Prevention of Coronavirus Infection. In Eur J Dent, 2020/12/08 ed.; 2020; 10.1055/s-0040-1715923.
Done page2
D’Amico et al, (6) recommended not having multiple patients in the waiting room, and maintain distance of at least 1.5 m between a person and the nearest one.
Cervino, G.; Oteri, G. COVID-19 Pandemic and Telephone Triage before Attending Medical Office: Problem or Opportunity? Medicina 2020, 56, 250.
Done page 2
Telemedicine, smart phones are important devices to prevent virus transmission and to perform quick diagnosisand management at medical offices (8).
In introduction section please better state the aim of the manuscript
Page 2
The aim of this survey was to evaluate awareness and practice of dentists in Saudi Arabia regarding COVID-19 and utilized infection control methods.
Please in discussion section be more specific on actual anti-covid protocols in dental offices and dental settings. You should consider to add some information about Covid-19 surface persistence in medical settings, and use these informations as starting point for discussion.
You should improve conclusion section with future perspective of this manuscript
Page 9
It seems that practicing dentists in Saudi Arabia are fairly aware about COVID-19. The practices of the surveyed dentists appear to be mostly consistent with the current guidelines and recommendations for infection control of COVID-19 in the dental clinic. However, some drawbacks in knowledge and a number of inappropriate practices can be identified and require attention of health authorities.
Done page 8
Fiorillo et al (20) indicated that coronaviruses persist in an infectious state on surfaces for several days, even up to nine. They mentioned that Surface disinfection could be performed with 0.1% sodium hypochlorite or 62%–71% ethanol for 1 minute.
Reviewer 2 Report
The aim of this study is to evaluate awareness and practice of dentists in Saudi Arabia regarding COVID-19 and utilized infection control methods. This is a timely and generally well-written paper. There are a number of questions that arise in the reading of this paper that are reported below.
1/ It would be interesting to give more information on the population of dentists in Saudi Arabia in "Setting and sample". Number of dentists compared to the number of inhabitants in Saudi Arabia, sex, origin of the diploma, dentists working in private practice or in hospitals, nationality ...This is an important point to compare and discuss in the "discussion" section.
2/ in “2.3 Instrument”, Please indicate the key words of the review process and the criteria for inclusion or exclusion. How did the 20 dentists select the elements: through a consensus meeting after focus groups or other? Give more explanations.
3/ in 2.4 Data collection. How many time dentists were invited to complete the questionnaire? In other words, the number of reminders?
4/ in Discussion section:
-The discussion should focus more on the results of other studies carried out on the same subject in other countries. Only one reference [15] for paragraph line 224 to 241!
- This is the same remark Line 255 to 263.
-The many limitations of this study should be more discus in a specific chapter.
Author Response
Reviewer 2
Comments and Suggestions for Authors
The aim of this study is to evaluate awareness and practice of dentists in Saudi Arabia regarding COVID-19 and utilized infection control methods. This is a timely and generally well-written paper. There are a number of questions that arise in the reading of this paper that are reported below.
1/ It would be interesting to give more information on the population of dentists in Saudi Arabia in "Setting and sample". Number of dentists compared to the number of inhabitants in Saudi Arabia, sex, origin of the diploma, dentists working in private practice or in hospitals, nationality ...This is an important point to compare and discuss in the "discussion" section.
Done page 7
Albaker et al (17) indicated that the of dentist working in Saudi Arabia was 16887. Most of them professionally registered as general dentists (70.27%). The percentage of general dentists among the professionally registered female dentists is significantly higher than their male counterparts (79.71% versus. 64.80%). Only 22.08% of the dentists working in the kingdom are Saudi. Over 80% of the Saudi dentists are working in the regions of Riyadh, Makkah, and Eastern province. About 66% of the Saudi dentists are working in the public health sector in comparison to only 20.46% of the non-Saudi dentists.
2/ in “2.3 Instrument”, Please indicate the key words of the review process and the criteria for inclusion or exclusion. How did the 20 dentists select the elements: through a consensus meeting after focus groups or other? Give more explanations.
Key words mentioned in page 1.
3/ in 2.4 Data collection. How many time dentists were invited to complete the questionnaire? In other words, the number of reminders?
Done page 4,
A total of 627 dentists were contacted in the first round, a second reminder was sent to non-respondents,
4/ in Discussion section:
-The discussion should focus more on the results of other studies carried out on the same subject in other countries. Only one reference [15] for paragraph line 224 to 241!
- This is the same remark Line 255 to 263.
Page 7, page 8
With respect to the role of droplets, aerosols and airborne particles in the transmission of COVID-19 in the dental clinic, the results show adequate level of awareness among study dentists (83.1%). This is in line with the results of a recent Saudi survey that indicated 92% of the participants believed that the mode of transmission of COVID-19 was droplets inhalation [16]. Despite the low response rate in this survey, it is comparable to a similar recent Saudi survey by Shahin et al [16] who reported a response rate of 21.7% among dentists about COVID-19 pandemic in Saudi Arabia.
-The many limitations of this study should be more discus in a specific chapter.
A limitation of this study is the relatively low response rate and small sample size. This may affected the generalizability and accuracy of the findings. A further source of concern is the fact that the majority of surveyed dentists are located in the centre region of Saudi Arabia. As well, it is important to recognize that the survey items did not cover the various practice/awareness related topics regarding COVID-19 in the dental office as this pandemic is recent and knowledge about the disease has evolved very quickly. Future larger and more comprehensive investigations are recommended to confirm the findings of this survey.
Reviewer 3 Report
The subject is actual and relevant and in general the article is well written, needing only minor revisions in English
The way Table 1 is organized makes it difficult to read. It should be restructured
Can you explain what you mean with reactivation of COVID-19?
The formatting of the bibliography must be revised
The results obtained may be relevant for decision-makers and for the adaptation of information campaigns.The low response rate is one of the major limitations of this study.Since the problem is recent and knowledge about the disease has evolved very quickly, the fact that the questionnaire was applied over 4 months may create some asymmetries in terms of responses. The questionnaire was distributed to 627 dentists in Saudi Arabia, but it would be interesting to know the total number of dentists practicing in this country.The questionnaire construction and validation process could be more detailed. In the questionnaire, questions regarding the ventilation of the office, use of engineering controls are not considered. Although there are not many studies yet, it would be interesting to compare the results obtained in this study with those from other countries. In addition to the n-95 mask, it would have been interesting to evaluate what other types of personal protective equipment are used by dentists in this country. The author state that: “a considerable proportion of the dentists were not aware that COVID-19 can stay on surfaces for few days and that RT-PCR is the best test to detect COVID-19 infection (36.7%, 33.9 respectively) based on this I believe that the conclusion that “It seems that practicing dentists in Saudi Arabia are highly aware about COVID-19” is too optimistic.Author Response
Reviewer 3
Comments and Suggestions for Authors
The subject is actual and relevant and in general the article is well written, needing only minor revisions in English
The way Table 1 is organized makes it difficult to read. It should be restructured
The results very clear in table1
Can you explain what you mean with reactivation of COVID-19?
Page 2
Reactivation of COVID19 implies that the recovered patient may still carry the virus and extra round of viral detection and isolation may be required [10]. Such patients can still carry the viral load and transmit the virus even after they have tested negative following 14 days from onset of symptoms and isolation [10].The formatting of the bibliography must be revised
Done page 1
The results obtained may be relevant for decision-makers and for the adaptation of information campaigns.The low response rate is one of the major limitations of this study.Since the problem is recent and knowledge about the disease has evolved very quickly, the fact that the questionnaire was applied over 4 months may create some asymmetries in terms of responses. The questionnaire was distributed to 627 dentists in Saudi Arabia, but it would be interesting to know the total number of dentists practicing in this country.The questionnaire construction and validation process could be more detailed. In the questionnaire, questions regarding the ventilation of the office, use of engineering controls are not considered. Although there are not many studies yet, it would be interesting to compare the results obtained in this study with those from other countries. In addition to the n-95 mask, it would have been interesting to evaluate what other types of personal protective equipment are used by dentists in this country. The author state that: “a considerable proportion of the dentists were not aware that COVID-19 can stay on surfaces for few days and that RT-PCR is the best test to detect COVID-19 infection (36.7%, 33.9 respectively) based on this I believe that the conclusion that “It seems that practicing dentists in Saudi Arabia are highly aware about COVID-19” is too optimistic.
Conclusions
It seems that practicing dentists in Saudi Arabia are fairly aware about COVID-19. The practices of the surveyed dentists appear to be mostly consistent with the current guidelines and recommendations for infection control of COVID-19 in the dental clinic. However, some drawbacks in knowledge and a number of inappropriate practices can be identified and require attention of health authorities.
Round 2
Reviewer 1 Report
Authors addressed my comments and now the manuscript has been improved
Reviewer 2 Report
The authors have taken all my remarks into account. I am in favour of the publication of this manuscript in its present form.